# Distribution of vitamin D status in the UK: a cross-sectional analysis of UK Biobank

Liang-Yu Lin , Liam Smeeth, Sinead Langan , Charlotte Warren-Gash

► Prepublication history and additional material are published online only. To view please visit the journal online (http://dx.doi.org/10.1136/bmjopen-2020-042305).

Department of Non-communicable Disease Epidemiology, London School of Hygiene and Tropical Medicine Faculty of Epidemiology and Population Health, London, UK

**Correspondence to**
Dr Liang-Yu Lin;
liang-yu.lin@lshtm.ac.uk

## ABSTRACT

**Objective** No recent large studies have described the distribution of vitamin D status in the UK. Understanding the epidemiology of vitamin D deficiency is important to inform targeted public health recommendations. This study aimed to investigate the distribution of factors associated with serum vitamin D status in a large national cohort.

**Design** A cross-sectional study.

**Setting** The UK Biobank, a prospective cohort study following the health and well-being of middle-aged and older adults recruited between 2006 and 2010.

**Participants** A total of 449 943 participants aged 40–69 years with measured serum vitamin D status were eligible for the analysis. Participants completed a questionnaire about sex, age, ethnic background, vitamin D supplementation, smoking, drinking and socioeconomic status.

**Primary and secondary outcome measures** We investigated the distribution of serum vitamin D status and the association between demographic factors and vitamin D deficiency or insufficiency. Vitamin D deficiency was defined as a serum 25-hydroxyvitamin D level <25 nmol/L. Multivariable logistic regression was used to assess the association between demographic factors and vitamin D status.

**Results** Asian (n=4297/8000, 53.7%) and black (n=2459/7046, 34.9%) participants had a higher proportion of vitamin D deficiency than white participants (n=50 920/422 907, 12%). During spring and winter, the proportion of vitamin D deficiency was higher across the UK and higher in the north than in the south. Male sex, abnormal body mass index, non-white ethnic backgrounds, smoking and being more socioeconomically deprived were associated with higher odds of vitamin D deficiency. Increasing age, taking vitamin D supplements and drinking alcohol were associated with lower odds of deficiency.

**Conclusions** Vitamin D status varied among different ethnic groups and by season and geographical area within the UK. Taking supplements was associated with a lower risk of vitamin D deficiency. These findings support the vitamin D supplementation recommendations of Public Health England.

## Strengths and limitations of this study

► This study used a large cohort of British adults in middle and older age, and its large sample size provides more statistical power in assessing associations.
► The findings provide further evidence to support the nutritional supplementation recommendations of Public Health England.
► The cohort is not nationally representative, so the findings cannot be extrapolated to the general population.
► The questionnaire on vitamin supplementation was self-reported and thus cannot be used to estimate the required supplement amount.

or ingestion, it is further metabolised in the liver into 25-hydroxyvitamin D (25(OH)D), which can be measured in the blood. 25(OH)D is further transformed in the kidneys into 1,25-dihydroxyvitamin D, the active form of vitamin D. This active vitamin D metabolite acts on the intestines and kidneys, regulates the absorption and excretion of calcium and phosphate, and facilitates the mineralisation of bone. Vitamin D deficiency may impair bone mineralisation, leading to osteopenia or osteoporosis.[1] Currently, Public Health England suggests that people older than 4 years of age should take 10 µg (400 IU) of vitamin D daily as a supplement during the winter to support musculoskeletal health.[2]

In addition to the classical effects of mineral homeostasis, more recent attention has focused on novel effects of vitamin D. Previous studies have indicated that vitamin D may have potential immunomodulatory effects. Active vitamin D can enhance innate immunity by increasing the production of antimicrobial peptides, and vitamin D can regulate adaptive immunity as well.[3] Epidemiological studies have also indicated that vitamin D is associated with autoimmune diseases such as inflammatory bowel disease, type I diabetes mellitus and multiple sclerosis.[4] In addition,

## INTRODUCTION

Vitamin D is essential to bone formation. It is mainly produced by the skin after sun exposure, and it can also be obtained from food and supplementation. After production

**Table 1** Comparison of included and excluded participants

| Characteristics | Included (N=449 943) | Excluded (N=52 550) |
|---|---|---|
| **Sex** | | |
| Female | 240 976 (53.6%) | 32 402 (61.7%) |
| Male | 208 967 (46.4%) | 20 148 (38.3%) |
| Average age (SD) | 56.5 (8.1) | 57.0 (7.8) |
| **BMI*** | | |
| Underweight | 2307 (0.5%) | 321 (0.6%) |
| Healthy weight | 145 082 (32.3%) | 15 222 (29.7%) |
| Overweight | 190 177 (42.4%) | 22 053 (43.1%) |
| Obese | 110 541 (24.7%) | 13 600 (26.6%) |
| **Central obesity*** | | |
| Not obese | 177 869 (39.6%) | 17 719 (34.5%) |
| Central obesity | 271 126 (60.4%) | 33 642 (65.5%) |
| **Ethnic background** | | |
| White | 424 213 (94.4%) | 47 901 (91.3%) |
| Mixed | 2593 (0.6%) | 316 (0.6%) |
| Asian | 8016 (1.8%) | 1823 (3.5%) |
| Black | 7050 (1.6%) | 984 (1.9%) |
| Chinese | 1407 (0.3%) | 167 (0.3%) |
| Other ethnicity† | 6068 (1.4%) | 1267 (2.4%) |
| **Tobacco smoking** | | |
| Non-smoker | 402 337 (89.6%) | 45 860 (88.3%) |
| Current smoker | 46 793 (10.4%) | 6184 (11.9%) |
| **Alcohol drinking status** | | |
| Never drink | 35 590 (7.9%) | 5071 (9.8%) |
| Drink occasionally | 101 690 (22.7%) | 12 208 (23.5%) |
| Drink weekly | 220 709 (49.2%) | 24 028 (46.2%) |
| Drink daily | 90 984 (20.3%) | 10 713 (20.6%) |
| **Vitamin and mineral supplementation use** | | |
| Vitamin D and associated mineral supplement‡ | 285 524 (63.6%) | 31 097 (63.9%) |
| Other vitamin and mineral supplement§ | 163 214 (36.4%) | 17 537 (36.1%) |
| Mean outdoor time in summer (SD) | 3.8 (2.4) | 3.8 (2.4) |
| Mean outdoor time in winter (SD) | 1.9 (1.8) | 1.9 (1.8) |
| **Index of Multiple Deprivation (IMD)¶** | | |
| 1 (Least deprived) | 88 516 (20.2%) | 9616 (18.7%) |
| 2 | 88 210 (20.1%) | 9782 (19.1%) |
| 3 | 87 584 (20.0%) | 10 249 (20.0%) |
| 4 | 87 585 (20%) | 10 306 (21.0%) |
| 5 (most deprived) | 86 539 (19.7%) | 11 370 (22.2%) |

*The classification is suggested by NICE guidelines.
†Includes any other unlisted ethnic groups, unclear ethnic groups and participants prefer not to answer.
‡Vitamin D and associated mineral: vitamin D, multivitamin, fish oil and calcium supplementation.
§Other vitamin and mineral supplements: vitamin A, B, C, E, folic acid or folate, glucosamine, zinc, iron, selenium and other supplements.
¶IMD scores were by quintile.
BMI, body mass index; NICE, National Institute for Health and Care Excellence.

a meta-analysis using original data from trials suggested that taking vitamin D supplementation may decrease the risk of respiratory infections.[5]

Although 1,25-dihydroxyvitamin D plays an active role in metabolism, its half-life is less than 4 hours, while the half-life of 25(OH)D is around 2–3 weeks. Thus, clinical 25(OH)D levels in the blood have been used to assess vitamin D status.[6] 25(OH)D can be analysed using either chemiluminescence immunoassay or tandem mass spectrometry, which are both recognised by the Royal Osteoporosis Society and Public Health England.[6 7] However, no current consensus exists about the definition of vitamin D deficiency, so each study may use different standards. The Endocrine Society of the USA defined vitamin D deficiency as 25(OH)D below 50 nmol/L, while the criterion of Public Health England is 25(OH)D less than 25 nmol/L.

Using the criterion of a blood vitamin D level less than 25 nmol/L, a cohort of British Caucasians (n=7437) indicated that in winter and spring, the average prevalence of vitamin D deficiency was 15.5% among people who were 45 years old.[8] Similarly, among adults in the National Diet and Nutrition Survey in the UK (n=3450), the prevalence of vitamin D deficiency was approximately 24.0% in men and 21.7% in women aged between 19 and 64 years.[9] However, these studies were relatively small, and no recent large study has described the distribution of vitamin D status in the UK.

The UK Biobank is a prospective, nationwide cohort used to investigate risk factors for major diseases in middle and old age; cohort participants were assessed for various biochemical biomarkers, including serum vitamin D status.[10 11] By using UK Biobank data, we aimed to conduct a cross-sectional study to investigate different demographic, seasonal and regional factors associated with vitamin D deficiency distribution in the UK.

## METHODS
### Study population
The UK Biobank was compiled from 2006 to 2010 by recruiting participants throughout the UK. People aged 40–69 years who lived within 40 km of 1 of the 22 UK Biobank assessment centres and who were registered with the UK National Health Service were eligible to be included in the cohort; approximately 500 000 volunteers were recruited.[10 12] UK Biobank participants received a wide range of examinations, including questionnaires and physical measures, and blood, urine and saliva samples were assayed.[6] Only participants with results for serum vitamin D status were included in our analyses.

### Measurement of covariates
At each participant's first visit to a UK Biobank assessment centre, a touchscreen questionnaire collected basic demographic characteristics; sociodemographic, environmental and lifestyle factors; and the date of assessment.[10] Information such as sex, age, ethnic background, skin

**Table 2** Basic characteristics of the study population

| Characteristics | Vitamin D deficiency (<25 nmol/L) | Vitamin D insufficiency (25–50 nmol/L) | Vitamin D sufficiency (>50 nmol/L) |
|---|---|---|---|
| Serum vitamin D levels (SD) (nmol/L) | 19.2 (3.9) | 37.8 (7.1) | 67.7 (14.5) |
| Sex | | | |
| Female (N=240 263) | 32 180 (13.4%) | 100 423 (41.8%) | 107 660 (44.8%) |
| Male (N=208 338) | 28 507 (13.7%) | 87 368 (41.9%) | 92 463 (44.4%) |
| Age (SD) | 54.8 (8.2) | 56.2 (8.1) | 57.3 (8.0) |
| BMI* | | | |
| Underweight (N=2301) | 423 (18.3%) | 844 (36.7%) | 1034 (45%) |
| Healthy weight (N=144 591) | 15 886 (11.0%) | 54 115 (37.4%) | 74 590 (51.6%) |
| Overweight (N=189 583) | 22 841 (12.1%) | 79 351 (41.9%) | 87 391 (46.1%) |
| Obese (N=110 292) | 20 990 (19.0%) | 52 704 (47.8%) | 36 598 (33.2%) |
| Central obesity* | | | |
| Non-obese (N=177 348) | 19 514 (11.0%) | 66 998 (37.8%) | 90 836 (51.2%) |
| Central obesity (N=270 306) | 40 903 (15.1%) | 120 389 (44.5%) | 109 014 (40.3%) |
| Ethnic background | | | |
| White (N=422 907) | 50 920 (12.0%) | 176 195 (41.7%) | 195 792 (46.3%) |
| Mixed (N=2589) | 642 (24.8%) | 1241 (47.9%) | 706 (27.3%) |
| Asian (N=8000) | 4297 (53.7%) | 2979 (37.3 %) | 724 (9.1%) |
| Black (N=7046) | 2459 (34.9%) | 3503 (49.7%) | 1084 (15.4%) |
| Chinese (N=1405) | 381 (27.1%) | 787 (56.0%) | 237 (16.9%) |
| Other ethnicity† (N=6059) | 1860 (30.7%) | 2816 (46.5%) | 1383 (22.8%) |
| Tobacco smoking | | | |
| Non-smoker (N=401 069) | 50 270 (12.5%) | 167 496 (41.8%) | 183 303 (45.7%) |
| Current smoker (N=46 721) | 10 170 (21.7%) | 19 944 (42.7%) | 16 607 (35.6%) |
| Alcohol drinking status | | | |
| Never drink (N=35 503) | 8419 (23.7%) | 15 265 (43.0%) | 11 819 (33.3%) |
| Drink occasionally (N=101 398) | 16 655 (16.4%) | 45 208 (44.6%) | 39 535 (39.0%) |
| Drink weekly (N=219 988) | 24 577 (11.1%) | 90 836 (41.3%) | 104 575 (47.6%) |
| Drink daily (N=90 742) | 10 730 (11.8%) | 36 063 (39.7%) | 43 949 (48.4%) |
| Mean outdoor time in summer (hours) (SD) | 3.3 (2.4) | 3.6 (2.4) | 4.1 (2.4) |
| Mean outdoor time in winter (hours) (SD) | 1.7 (1.8) | 1.9 (1.8) | 2.1 (1.8) |
| Vitamin and mineral supplementation use | | | |
| Other vitamin or mineral supplement‡ (N=284 768) | 48 877 (17.1%) | 128 478 (45.1%) | 107 413 (37.7%) |
| Vitamin D and associated mineral supplement§ (N=162 629) | 11 368 (7.0%) | 58 801 (36.2%) | 92 460 (56.9%) |
| Index of Multiple Deprivation (IMD)¶ | | | |
| 1 (least deprived, N=88 264) | 8414 (9.5%) | 34 962 (39.6%) | 44 888 (50.9%) |
| 2 (N=87 909) | 9321 (10.6%) | 36 073 (41.0%) | 42 515 (48.4%) |
| 3 (N=87 268) | 10 480 (12.0%) | 36 049 (41.3%) | 40 739 (46.7%) |
| 4 (N=87 321) | 13 211 (15.1%) | 37 646 (43.1%) | 36 464 (41.8%) |
| 5 (most deprived, N=86 347) | 17 602 (20.4%) | 38 369 (44.4%) | 30 376 (35.2%) |
| Seasons | | | |
| Spring (N=129 570) | 25 912 (20%) | 62 378 (48.1%) | 41 280 (31.7%) |
| Summer (N=118 924) | 5385 (4.5%) | 39 856 (33.5%) | 73 683 (62.0%) |
| Autumn (N=108 888) | 8324 (7.6%) | 41 496 (38.1%) | 59 068 (54.3%) |
| Winter (N=91 219) | 21 066 (23.1%) | 44 061 (48.3%) | 26 092 (28.6%) |

Continued

**Table 2** Continued

| Characteristics | Vitamin D deficiency (<25 nmol/L) | Vitamin D insufficiency (25–50 nmol/L) | Vitamin D sufficiency (>50 nmol/L) |
|---|---|---|---|
| Regions of UK Biobank assessment centres | | | |
| South West (N=38 872) | 3068 (7.9%) | 14 622 (37.6%) | 21 182 (54.5%) |
| South East (N=39 814) | 3245 (8.2%) | 15 400 (38.7%) | 21 169 (53.2%) |
| London (N=61 291) | 10 232 (16.7%) | 27 037 (44.1%) | 24 022 (39.2%) |
| East Midlands (N=30 337) | 3001 (9.9%) | 12 115 (39.9%) | 15 221 (50.2%) |
| West Midlands (N=40 044) | 6785 (17.0%) | 17 670 (44.1%) | 15 589 (38.9%) |
| Wales (N=19 142) | 2732 (14.3%) | 8808 (46.0%) | 7614 (39.8%) |
| Yorkshire and The Humber (N=66 197) | 8372 (12.7%) | 27 878 (42.1%) | 29 947 (45.2%) |
| North West (N=68 196) | 8715 (12.8%) | 27 924 (41.0%) | 31 557 (46.3%) |
| North East (N=52 277) | 6919 (13.2%) | 21 174 (40.5%) | 24 184 (46.3%) |
| Scotland (N=32 419) | 7618 (23.5%) | 15 163 (46.8%) | 9638 (29.7%) |

*The classification is suggested by NICE guidelines.
†Includes any other unlisted ethnic groups, unclear ethnic groups and participants prefer not to answer.
‡Vitamin D and associated mineral: vitamin D, multivitamin, fish oil and calcium supplementation.
§Other vitamin and mineral supplements: vitamin A, B, C, E, folic acid or folate, glucosamine, zinc, iron, selenium and other supplements.
¶IMD scores were by quintile.
BMI, body mass index; NICE, National Institute for Health and Care Excellence.

colour, smoking status, drinking status, sun exposure, and vitamin and mineral supplement use were recorded. Physical measurements, including body mass index (BMI) and waist circumference were also taken. The Index of Multiple Deprivation (IMD) of participants was obtained through data linkage.[13]

### Defining serum vitamin D status

Biochemical assays were performed on blood samples collected during the baseline evaluation at the assessment centres.[6] Briefly, serum samples were collected in a silica clot accelerator tube and stored at −80°C. These samples were later processed in a central laboratory using an automated dispensing system.[14] Serum 25(OH)D status was measured by chemiluminescence immunoassay (DiaSorin LIAISON XL, Italy), which was certified by the Vitamin D Standardization-Certification Program of the Centers for Disease Control and Prevention.[15] To ensure the precision of analysis, quality control samples at different concentrations were analysed,[16] and the testing assay for vitamin D was verified through the RIQAS Immunoassay Speciality I EQA programme (Randox Laboratories), an external quality assurance scheme.[17]

### Statistical analysis

This was a cross-sectional study describing the distribution of serum vitamin D status, so only participants who had at least one available measurement of serum vitamin D status were included. We used the standards of vitamin D deficiency adopted by Public Health England.[6 18] A serum 25(OH)D level less than 25 nmol/L was coded as 'deficiency', and 25–50 nmol/L was coded as 'insufficiency'. A 25(OH)D level greater than 50 nmol/L was coded as

'sufficiency'. Vitamin D was coded as missing in cases of no reportable value, values above or below the reportable limits, or unrecoverable aliquot problems.

To compare the distribution of vitamin D status by these factors associated with vitamin D, continuous covariates, such as BMI and waist circumference, were coded as categorical variables following National Institute for Health and Care Excellence guidelines.[19] According to the guidelines, different BMI and waist circumference standards were applied for Asian and Chinese populations. Self-reported ethnic backgrounds were coded into six groups according to the original questionnaire. Participants with any other unlisted ethnic group, with an unclear ethnic group or preferring not to answer were assigned 'other ethnic group'. The mean outdoor time was recorded as 0.5 hours/day if it was less than 1 hour. Responses regarding vitamin and mineral use were further regrouped as 'vitamin D, multivitamin, fish oil and calcium use' and 'other vitamin and mineral use'. IMD scores were categorised into five quintiles, with the fifth quintile assigned the 'most deprived' group. The locations of 22 Biobank assessment centres were grouped using the geographical regions of the UK.[20] The date of blood collection for vitamin D examination was categorised into four seasons, according to the Met Office definitions.[21]

Demographic factors including sex, age (modelled as a continuous variable), ethnicity, BMI, smoking, drinking alcohol, IMD, vitamin D testing season and geographical location were further analysed in terms of their association with vitamin D insufficiency (<50 nmol/L) and deficiency (<25 nmol/L) using simple and multivariable

**Table 3** The association between demographic characteristics and low vitamin D status

| Characteristics | OR of vitamin D insufficiency (25(OH)D <50 nmol/L) | | OR of vitamin D deficiency (25(OH)D <25 nmol/L) | |
| --- | --- | --- | --- | --- |
| | Crude | Adjusted* | Crude | Adjusted* |
| Sex | | | | |
| Female | 1 | 1 | 1 | 1 |
| Male | 1.02 (1.0, 1.03) | 0.91 (0.90, 0.93) | 1.02 (1.01, 1.04) | 0.91 (0.9, 0.93) |
| Age (SD) | 0.98 (0.97, 0.97) | 0.98 (0.98, 0.98) | 0.97 (0.96, 0.97) | 0.98 (0.98, 0.98) |
| BMI | | | | |
| Healthy weight | 1 | 1 | 1 | 1 |
| Underweight | 1.3 (1.20, 1.42) | 1.26 (1.14, 1.38) | 1.83 (1.64, 2.03) | 1.71 (1.51, 1.93) |
| Overweight | 1.25 (1.23, 1.26) | 1.24 (1.22, 1.25) | 1.11 (1.09, 1.13) | 1.04 (1.01, 1.06) |
| Obese | 2.15 (2.11, 2.18) | 2.08 (2.04, 2.12) | 1.91 (1.86, 1.95) | 1.68 (1.64, 1.72) |
| Ethnic background | | | | |
| White | 1 | 1 | 1 | 1 |
| Mixed | 2.30 (2.11, 2.51) | 2.24 (2.03, 2.46) | 2.42 (2.21. 2.64) | 2.31 (2.09, 2.56) |
| Asian | 8.67 (8.02, 9.35) | 8.54 (7.87, 9.27) | 8.47 (8.10, 8.86) | 10.99 (10.39, 11.62) |
| Black | 4.74 (4.44, 5.06) | 4.14 (3.85, 4.45) | 3.93 (3.74, 4.13) | 3.6 (3.39, 3.83) |
| Chinese | 4.25 (3.70, 4.89) | 4.42 (3.81, 5.14) | 2.72 (2.42, 3.06) | 2.77 (2.42, 3.18) |
| Other ethnicity | 2.91 (2.74, 3.10) | 2.73 (2.55, 2.93) | 3.24 (3.07, 3.42) | 3.11 (2.9, 3.33) |
| Tobacco smoking | | | | |
| Non-smoker | 1 | 1 | 1 | 1 |
| Current smoker | 1.53 (1.50, 1.56) | 1.43 (1.40, 1.46) | 1.94 (1.90, 1.99) | 1.82 (1.77, 1.87) |
| Alcohol drinking status | | | | |
| Never drink | 1 | 1 | 1 | 1 |
| Drink occasionally | 0.78 (0.76, 0.8) | 0.85 (0.84, 0.88) | 0.63 (0.61, 0.65) | 0.75 (0.73, 0.78) |
| Drink weekly | 0.55 (0.53, 0.56) | 0.66 (0.64, 0.68) | 0.40 (0.39, 0.42) | 0.55 (0.53, 0.57) |
| Drink daily | 0.53 (0.52, 0.55) | 0.70 (0.68, 0.72) | 0.43 (0.42, 0.45) | 0.66 (0.64, 0.69) |
| Vitamin and mineral supplementation | | | | |
| Other vitamin and mineral supplement | 1 | 1 | 1 | 1 |
| Vitamin D, multivitamin, fish oil and calcium supplement | 0.46 (0.45, 0.47) | 0.41 (0.41, 0.42) | 0.36 (0.35, 0.37) | 0.32 (0.31, 0.33) |
| Index of Multiple Deprivation (IMD)† | | | | |
| 1 (least deprived) | 1 | 1 | 1 | 1 |
| 2 | 1.11 (1.09, 1.13) | 1.03 (1.01, 1.05) | 1.12 (1.09, 1.16) | 1.02 (0.98, 1.05) |
| 3 | 1.18 (1.16, 1.2) | 1.05 (1.03, 1.07) | 1.29 (1.26, 1.33) | 1.11 (1.07, 1.14) |
| 4 | 1.44 (1.42, 1.47) | 1.17 (1.14, 1.19) | 1.69 (1.64, 1.74) | 1.27 (1.23, 1.31) |
| 5 (most deprived) | 1.91 (1.87, 1.94) | 1.34 (1.31, 1.37) | 2.43 (2.36, 2.50) | 1.53 (1.48, 1.58) |
| Seasons | | | | |
| Summer | 1 | 1 | 1 | 1 |
| Spring | 3.48 (3.42, 3.54) | 3.86 (3.79, 3.93) | 5.26 (5.1, 5.42) | 6.43 (6.22, 6.65) |
| Autumn | 1.37 (1.35, 1.40) | 1.43 (1.40, 1.45) | 1.74 (1.68, 1.80) | 1.89 (1.82, 1.96) |
| Winter | 4.06 (3.99, 4.14) | 4.56 (4.47, 4.65) | 6.31 (6.12, 6.51) | 7.82 (7.55, 8.1) |
| Regions (categorised centres) | | | | |
| South West | 1 | 1 | 1 | 1 |
| South East | 1.05 (1.03, 1.08) | 1.15 (1.11, 1.18) | 1.04 (0.98, 1.09) | 1.11 (1.05, 1.18) |
| London | 1.86 (1.81, 1.91) | 1.31 (1.28, 1.35) | 2.34 (2.24, 2.44) | 1.31 (1.25, 1.38) |
| East Midlands | 1.19 (1.15, 1.23) | 1.07 (1.04, 1.11) | 1.28 (1.22, 1.35) | 1.13 (1.07, 1.2) |
| West Midlands | 1.88 (1.82, 1.93) | 1.22 (1.18, 1.26) | 2.38 (2.27, 2.48) | 1.26 (1.19, 1.32) |

Continued

**Table 3** Continued

| Characteristics | OR of vitamin D insufficiency (25(OH)D <50 nmol/L) | | OR of vitamin D deficiency (25(OH)D <25 nmol/L) | |
|---|---|---|---|---|
| | Crude | Adjusted* | Crude | Adjusted* |
| Wales | 1.81 (1.75, 1.88) | 1.03 (0.99, 1.07) | 1.94 (1.84, 2.05) | 1.03 (0.97, 1.09) |
| Yorkshire and The Humber | 1.45 (1.41, 1.48) | 1.19 (1.16, 1.23) | 1.68 (1.61, 1.76) | 1.30 (1.24, 1.36) |
| North West | 1.38 (1.35, 1.42) | 1.15 (1.12, 1.18) | 1.68 (1.61, 1.76) | 1.28 (1.22, 1.34) |
| North East | 1.39 (1.35, 1.43) | 1.23 (1.19, 1.27) | 1.78 (1.7, 1.86) | 1.53 (1.46, 1.61) |
| Scotland | 2.83 (2.74, 2.92) | 1.98 (1.91, 2.05) | 3.58 (3.43, 3.75) | 2.39 (2.28, 2.51) |

*Adjusted for sex, age, BMI categories, ethnicity, smoking, drinking, vitamin D supplementation, IMD, seasons, regions of UK Biobank assessment centres.
†IMD scores were by quintile.
BMI, body mass index; 25(OH)D, 25-hydroxyvitamin D.

logistic regression models. Covariates such as central obesity were not included in the model to avoid probable collinearity with BMI. All statistical analyses were carried out using Stata/MP V.15 (StataCorp, USA).

### Patient and public involvement
Patients and/or the public were not involved in this retrospective study.

### RESULTS
#### Comparison of included and excluded participants
We compared the included participants with those who did not have their vitamin D status tested (table 1). Vitamin D values were missing for 52 550 participants, who were excluded. Among excluded participants, the proportion of female participants (61.7%) was slightly higher than among the included participants (53.6%), while the distribution of the other categorical variables was similar across the included and excluded groups. The average age and mean outdoor time in summer were similar among included and excluded participants, while included participants who spent more time outside in winter.

#### Demographic distribution of vitamin D status
The demographic characteristics according to vitamin D status are shown in table 2. The average vitamin D levels were 19.2 nmol/L in the deficient group, 37.8 nmol/L in the insufficient group and 67.7 nmol/L in the sufficient group. The distribution of vitamin D status was similar in both sexes, and the mean age of the vitamin D sufficiency group was older than other groups. The proportion of vitamin deficiency was lower among people with healthy weight (n=15 886, 11%) and non-obese people (n=177 348, 11%). Asian (n=4297, 53.7%) and black (n=2459, 34.9%) people had a higher proportion of vitamin D deficiency than white participants (n=50 920, 12.0%). Vitamin D deficiency was more prevalent among current smokers (n=10 170, 21.7%) and those who never drank alcohol (n=8419, 23.7%). People deficient in vitamin D spent less outdoor time in winter (mean=1.7 hours). The proportion

of vitamin D deficiency was lower among participants taking vitamin D and associated mineral supplements (n=11 368, 7%) compared with people taking any other supplements (n=48 877, 17%). The most deprived people had the highest proportion of vitamin D deficiency (n=17 602, 20.4%), while the least deprived participants had the lowest deficiency proportion (n=8414, 9.5%).

Vitamin D deficiency was more common in winter (n=21 066, 23.1%) and spring (25 912, 20%). Participants who visited assessment centres in Scotland had the highest proportion of vitamin D deficiency (n=7618, 23.5%) compared with other assessment centres in the southern UK (table 2). A map showing the distribution of the proportion of vitamin D deficiency in different seasons by each UK Biobank assessment centre is illustrated in online supplemental figure 1. This shows that in winter and spring, vitamin D deficiency was prevalent across the country, while in summer and autumn it was less common. Furthermore, in spring, autumn and winter, the proportion of vitamin D deficiency in the northern part of the UK was higher than in the southern part of the country (online supplemental figure 1).

#### Association between demographic factors and vitamin D status
To assess the association of factors related to vitamin D status, the ORs of having vitamin D deficiency (25(OH)D <25 nmol/L) or insufficiency (25(OH)D <50 nmol/L) were summarised in table 3. Male sex, abnormal BMI, smoking, non-white ethnicity, and being more deprived had greater odds of vitamin D deficiency or insufficiency. Increasing age, drinking alcohol, and taking vitamin D, multivitamin, fish oil or calcium supplements were associated with lower odds of vitamin D deficiency or insufficiency. Compared with summer, receiving testing in spring or winter had greater odds of vitamin D deficiency or insufficiency. For the regions with assessment centres, Scotland had higher odds of vitamin D deficiency and insufficiency compared with other regions in the UK. After adjusting for the variables listed in table 3, these associations remained (table 3).

## DISCUSSION

In this cross-sectional study, we found that vitamin D deficiency was more prevalent among people of colour, in spring and winter, and in the northern UK. Male sex, abnormal BMI, non-white ethnic background, smoking and being more deprived increased the odds of vitamin D deficiency. Taking vitamin D supplements, drinking alcohol and increasing age were associated with decreased odds of vitamin D deficiency.

Our study showed that vitamin D status was strongly associated with seasonality and geographical distribution, which is similar to previous studies.[8 9] Other studies have also indicated that vitamin D deficiency is more prevalent among non-white ethnic groups, such as south Asians. A small longitudinal study (n=140) in the UK showed that more than 90% of south Asian women in the cohort had vitamin D insufficiency (25(OH)D <50 nmol/L) throughout the year, while the proportion of insufficiency was much lower in the white comparison groups (20.3% in summer and 70% in winter).[22] A systematic review of 19 cross-sectional studies and 1 cohort study indicated that serum hydroxyvitamin D levels were lower among black, African American and non-Hispanic black groups, compared with Caucasian groups.[23] These findings indicate the importance of differences in ethnicity, latitude and season, which should always be considered in dietary guidelines and future nutritional studies of vitamin D.

We found that abnormal BMI, tobacco smoking, and deprivation increased the odds of vitamin D deficiency or insufficiency, while taking vitamin D and associated mineral supplements was associated with a lower risk of vitamin D deficiency or insufficiency. These results are consistent with other research about factors associated with vitamin D.[1 24 25] The common vitamin D supplements contain around 400–2000 IU of vitamin $D_2$ or $D_3$, meeting the official daily intake guide. Our findings provide some evidence about the effect of vitamin D supplements as well as the Public Health England recommendation for vitamin D supplementation in winter and for some ethnic backgrounds.[2]

Curiously, age and drinking alcohol were inversely associated with vitamin D status. Ageing has been regarded as a risk factor for vitamin D deficiency; however, our results showed that increasing age slightly decreased the odds of deficiency. Nevertheless, the mean age of our study population was 56.5 years, and age-related vitamin D deficiency tends to manifest at a more advanced age.[26] Another possible explanation is that our study population focused on middle-aged or older participants, who may be more likely to receive prescriptions containing vitamin D. However, this information was not included in our study. Regarding drinking alcohol, although a previous systematic review reported no consistent correlation between alcohol intake and serum vitamin D levels,[27] recent studies have shown a negative association. A study using the National Health and Nutrition Examination Survey database in the USA from 2001 to 2010 showed the prevalence of vitamin D deficiency among current alcohol drinkers was 38% lower than in non-drinkers.[25] Similarly, a cross-sectional survey in Portugal among 1500 participants aged over 65 years showed that alcohol drinkers had lower odds of vitamin D deficiency compared with non-drinkers (moderate drinker: OR=0.49 (CI: 0.32 to 0.73); excessive drinker: OR=0.48 (CI: 0.27 to 0.85)).[24] The mechanism behind this possible association is still unclear, and future studies about vitamin D should take alcohol into account.

The key strengths of our study are its large sample size and wide range of measurements, which provide more statistical power in analysing the factors associated with vitamin D deficiency. Moreover, information about the exposure and outcomes was collected following a predefined protocol, and samples were processed systematically, which minimises potential differential or non-differential misclassification bias. Our analysis about the association between demographic factors and vitamin D deficiency is solid. However, several important limitations need to be considered. Despite the large size of the UK Biobank, this cohort is not nationally representative. Compared with non-participants, the participants of the UK Biobank were more likely to be older, women, and with a higher socioeconomic status and fewer health conditions.[12] Due to this healthy volunteer effect, our findings on vitamin D status cannot be generalised to the UK population, and we may have underestimated the prevalence of vitamin D deficiency. Nevertheless, as the healthy volunteer effect does not affect the validity of exposure–outcome relationships, our findings on the factors associated with vitamin D status are likely to be valid.

This analysis has concentrated on vitamin D status using serum vitamin D levels obtained from a single blood test, which may not reflect long-term vitamin D status. Additionally, the questionnaires collecting information about vitamin D supplementation and outdoor physical activity could not precisely quantify the exposure. In further research, a more representative sample should be considered to assess the prevalence of vitamin D deficiency in the UK. Researchers may consider using repeated measurements to assess long-term vitamin D status more precisely or including clinical diagnosis or vitamin D prescription to capture clinical status. In addition, collecting information to quantify vitamin D supplementation and outdoor activities in different seasons can further validate the recommendation of Public Health England.

## CONCLUSION

Vitamin D deficiency was more common in winter and spring, and its prevalence was higher in the northern UK than the southern UK. Male sex, abnormal BMI, Asian and black ethnic backgrounds, and tobacco smoking were associated with higher odds of vitamin D deficiency. Taking vitamin D supplements and drinking alcohol were associated with lower odds of vitamin D deficiency. These results provide some evidence supporting the Public

Health England recommendation for taking vitamin D supplementation in winter and for people with black or Asian ethnic backgrounds.

**Contributors** L-YL contributed to the design and analysis of the study, drafted and revised the manuscript according to other authors' comments. CW-G contributed to the design of the study, made critical comments on the manuscript and revised the paper critically. SL contributed to the design of the study, made critical comments on the manuscript and revised the paper critically. LS contributed to the design of the study and revised the paper critically. All authors approved the final version of the manuscript.

**Funding** L-YL is funded by the scholarship of government sponsorship for overseas study by the Ministry of Education Republic of China (Taiwan). CW-G is supported by a Wellcome Intermediate Clinical Fellowship (201440_Z_16_Z). SL is funded by a Wellcome Senior Clinical Fellowship in Science (205039/Z/16/Z).

**Map disclaimer** The depiction of boundaries on this map does not imply the expression of any opinion whatsoever on the part of BMJ (or any member of its group) concerning the legal status of any country, territory, jurisdiction or area or of its authorities. This map is provided without any warranty of any kind, either express or implied.

**Competing interests** None declared.

**Patient consent for publication** Not required.

**Ethics approval** The UK Biobank obtained ethics approval from its Research Ethics Committee (reference: 11/NW/0382). Our study received ethics approval from the UK Biobank (ID:51265) and the Research Ethics Committee of the London School of Hygiene and Tropical Medicine (reference: 17158). Our study followed the principles of the Declaration of Helsinki.

**Provenance and peer review** Not commissioned; externally peer reviewed.

**Data availability statement** Data are available in a public, open access repository. Other researchers can apply for UK Biobank data to answer specific research questions. We will upload our analysis codes to LSHTM Data Compass, an open data repository for research outputs from LSHTM (https://datacompass.lshtm.ac.uk/).

**ORCID iDs**
Liang-Yu Lin http://orcid.org/0000-0003-4720-6738
Sinead Langan http://orcid.org/0000-0002-7022-7441
Charlotte Warren-Gash http://orcid.org/0000-0003-4524-3180

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
