## [Reviewer comments · BMJ Open]

ARTICLE DETAILS

TITLE (PROVISIONAL)	The distribution of Vitamin D status in the UK: a cross-sectional analysis of UK Biobank
AUTHORS	Lin, Liang-Yu; Smeeth, Liam; Langan, Sinead; Warren-Gash, Charlotte

VERSION 1 – REVIEW

REVIEWER	Emeritus Professor John Wark University of Melbourne, Australia
REVIEW RETURNED	02-Apr-2020

GENERAL COMMENTS	This study certainly has a worthwhile objective. However, there are important methodological shortcomings and apparently unjustified interpretations of the findings as presented. Throughout, describing serum 25-hydroxyvitamin D levels as vitamin D levels is inaccurate. The Abstract should briefly describe the UK Biobank. The findings cannot support any specific vitamin D supplementation recommendation when no vitamin D dosage data are provided and about 10% of the deficient group and 40% of the insufficient group reported using supplements; and supplement use is described as poorly reported. In the Introduction, the effects of vitamin D are poorly explained. There clearly was a healthy volunteer effect in recruitment of the UK Biobank cohort (see ref 7), but the implications of this limitation are poorly discussed. For example, the prevalence of vitamin D deficiency/insufficiency may well be underestimated. On the other hand, findings re associations with vitamin D status may be quite valid. Re Methods, performance criteria of the 25-OHD assay as used are not provided. It is stated that "values outside the reportable range" were excluded. Does this mean that participants with undetectable values were excluded? This needs clarification including reporting how many were excluded on this criterion. There is no statistical analysis of between group significance of associations with putative determinants of vitamin D status: this is a major deficiency in reporting the findings. For example, were the associations of vitamin D deficiency/insufficiency with obesity, Asian, Chinese and black ethnicity, smoking, never drinking statistically-significant? It is unfortunate that the relationship between comorbidities (e.g., as in ref 7) and vitamin D status was not investigated. In the Discussion, it is claimed that a larger sample size gives a "more accurate distribution of vitamin D status". Obviously this assertion is subject to inherent biases and methodological issues.
---

REVIEWER	Martin Hewison University of Birmingham, UK
REVIEW RETURNED	21-Aug-2020

GENERAL COMMENTS	The manuscript by Lin et al is a simple but well-written and effective statement on the prevalence of vitamin D-deficiency in the UK. By using UK Biobank data, the authors have shown clearly the low levels of vitamin D in some UK communities. This is further underlined by the fact that the authors used a 25-hydroxyvitamin D (25D) level of <25 nmol/L (10 ng/ml) as deficiency, despite the fact that many other studies use <50 nmol/L. Nevertheless, the authors have used 25-50 nmol/L 25D as one of the analysis groups, providing additional data. The resulting data strongly underline the advice by UK SACN for vitamin D supplementation on a national basis, particularly in ethnic groups within the UK. However, further analysis of the data is very limited and this reviewer felt that some important messages were missing. Specific comments 1) The serum 25D measurements were carried out using a chemiluminescence system that is no longer considered to be the gold-standard for analysis of 25D (liquid chromatography- tandem mass spectrometry) is now accepted as the definitive method for accurate vitamin D analysis) The author must provide detailed information on the reproducibility – intra-assay variation and DEQAS data for the assay that was used. 2) I may be missing something but there does not appear to be any statistical analysis. There appear to be some interesting variations in Table 2. For example, obese people are far less likely to be vitamin D-sufficient. Likewise, daily drinkers are far more likely to be vitamin D-sufficient but is this because this includes more of the Caucasian cohort? Trends need to be statistically defined – e.g. how much more time outdoors is required to produce vitamin D-sufficiency. At present the study is simply a list of data, there is no attempt to analyse these data. 3) The authors show seasonal variations in 25D levels but this is shown in a very user-unfriendly fashion. Most of the UK map is white and not part of the study. Why not just list the locations (maybe South to North or vice versa) for each season and show the actual serum 25 levels? 4) 'olive' colored skin appears to be more commonly associated with vitamin D-sufficiency. Can the authors comment on this? What are the other known features of this group. 5) Are any data for exercise levels available for the Biobank cohort?
--

VERSION 1 – AUTHOR RESPONSE

Reviewer 1 / comment 1:

Throughout, describing serum 25-hydroxyvitamin D levels as vitamin D levels is inaccurate.

Response:

We agree with the referee that describing serum 25-hydroxyvitamin D levels as vitamin D levels is not accurate, but this is the best available vitamin D metabolite that can be detected in the blood. To clarify this, we revised the third paragraph of introduction as below:

Although 1,25-dihydroxyvitamin D plays an active role in metabolism, its half-life is less than 4 hours, while the half-life of 25-hydroxyvitamin D is around 2–3 weeks. Thus, clinical 25-hydroxyvitamin D levels in the blood have been used to assess vitamin D status⁶.

We now also use the term 'vitamin D status' rather than 'levels' throughout the paper, with vitamin D deficiency and insufficiency states defined using Public Health England's definitions.

Reviewer 1 / comment 2:

The Abstract should briefly describe the UK Biobank.

Response:

In accordance with the referees' wishes, we have now changed the sentence of study setting in the abstract as below:

Setting: The UK Biobank, a prospective cohort study following the health and wellbeing of middle-aged and older adults recruited between 2006 and 2010.

Reviewer 1 / comment 3:

The findings cannot support any specific vitamin D supplementation recommendation when no vitamin D dosage data are provided and about 10% of the deficient group and 40% of the insufficient group reported using supplements; and supplement use is described as poorly reported. In the Introduction, the effects of vitamin D are poorly explained.

Response:

In our previous manuscript, we analyzed the vitamin D supplementation using 24-hour food recall questionnaires from the UK Biobank, which was only available for a small portion of our study participants. Following the reviewer's comments, we applied for new variables containing information about vitamin D and mineral supplementation for the whole cohort (N = 447,397), which were recorded when participants visited the UK Biobank assessment centres. The updated data are shown in Table 2:

Characteristics	Vitamin D deficiency (< 25 nmol/L)
Vitamin and mineral supplementation use	
Vitamin D and associated mineral supplement ¹ (N=284,768)	48,877
Other vitamin and mineral supplement (N=162,629)	11,368

1. Including vitamin D, multi-vitamin, fish oil (including cod liver oil), or calcium supplement

Regarding the effect of vitamin D, as well as the effects of vitamin D supplementation, we have revised the introduction and discussion as follows:

Introduction:

25-hydroxyvitamin D is further transformed in the kidneys into 1,25-dihydroxyvitamin D, the active form of vitamin D. This active vitamin D metabolite acts on the intestines and kidneys, regulates the absorption and excretion of calcium and phosphate, and facilitates the mineralisation of bone. Vitamin D deficiency may impair bone mineralisation, leading to osteopenia or osteoporosis¹. Currently, Public Health England suggests that people older than 4 years of age should take 10 μ g (400 IU) of vitamin D daily as a supplement during the winter to support musculoskeletal health².

In addition to the classical effects of mineral homeostasis, more recent attention has focused on novel effects of vitamin D. Previous studies have indicated that vitamin D may have potential immunomodulatory effects. Active vitamin D can enhance innate immunity by increasing the production of antimicrobial peptides, and vitamin D can regulate adaptive immunity as well³.

Epidemiological studies have also indicated that vitamin D is associated with autoimmune diseases such as inflammatory bowel disease, type I diabetes mellitus and multiple sclerosis⁴. In addition, a meta-analysis using original data from trials suggested that taking vitamin D supplementation may decrease the risk of respiratory infections⁵.

Although 1,25-dihydroxyvitamin D plays an active role in metabolism, its half-life is less than 4 hours, while the half-life of 25-hydroxyvitamin D is around 2–3 weeks. Thus, clinical 25-hydroxyvitamin D levels in the blood have been used to assess vitamin D status⁶. 25-hydroxyvitamin D can be analysed using either chemiluminescence immunoassay or tandem mass spectrometry, which are recognised by the Royal Osteoporosis Society and Public Health England^{6,7}. However, no current consensus exists about the definition of vitamin D deficiency, so each study may use different standards. The Endocrine Society of the US defined vitamin D deficiency as 25-hydroxyvitamin D below 50 nmol/L, while the criterion of Public Health England is 25-hydroxyvitamin D less than 25 nmol/L.

Discussion:

These results are consistent with other research about factors associated with vitamin D^{1 25 26}. The common vitamin D supplements contain around 400 to 2,000 IU of vitamin D₂ or D₃, meeting the official daily intake guide. Our findings provide some evidence about the effect of vitamin D supplements as well as the Public Health England recommendation for vitamin D supplementation in winter and for some ethnic backgrounds².

Reviewer 1 / comment 4:

There clearly was a healthy volunteer effect in recruitment of the UK Biobank cohort (see ref 7), but the implications of this limitation are poorly discussed. For example, the prevalence of vitamin D deficiency/insufficiency may well be underestimated. On the other hand, findings re associations with vitamin D status may be quite valid.

Response:

We now discuss the implications of the healthy volunteer effect for our findings in more detail. We revised the last paragraph of discussion as below:

Despite the large size of the UK Biobank, this cohort is not nationally representative. Compared with non-participants, the participants of the UK Biobank were more likely to be older, female, and with a higher socioeconomic status and fewer health conditions¹². Due to this healthy volunteer effect, our findings on vitamin D status cannot be generalised to the UK population, and we may have underestimated the prevalence of vitamin D deficiency. Nevertheless, as the healthy volunteer effect does not affect the validity of exposure–outcome relationships, our findings on the factors associated with vitamin D status are likely to be valid.

Reviewer 1 / comment 5:

Re Methods, performance criteria of the 25-OHD assay as used are not provided. It is stated that "values outside the reportable range" were excluded. Does this mean that participants with undetectable values were excluded? This needs clarification including reporting how many were excluded on this criterion.

Response:

We agree with the referee that the data collection process needs more clarification. The results of the biochemical assay were assigned as missing if there was no reportable data, which affected about 9% of the total assay results. The reasons for missing assay data were summarized as below:

Vitamin D missing reasons (N=45,588) *	
No data returned	19,499 (42.8%)
Original value above or below reportable limit	2,657 (5.8%)
Unrecoverable aliquot due to sample dilution problems	3,067 (6.7%)
Not enough sample results for accurate assessment of the dilution problems	668 (1.4%)
Analyzer deemed result not reportable for reason other than above or below reportable range	20,116 (44.1%)
Not reportable because dilution factor <0.9 or >1.1	28 (0.006%)

* Source: UK Biobank Data-Field 30895 - Vitamin D missing reason <http://biobank.ndph.ox.ac.uk/showcase/field.cgi?id=30895>

The text in the methods was revised as below:

"Vitamin D was coded as missing in cases of no reportable value, values above or below the reportable limits, or unrecoverable aliquot problems."

Reviewer 1 / comment 6

There is no statistical analysis of between group significance of associations with putative determinants of vitamin D status: this is a major deficiency in reporting the findings. For example, were the associations of vitamin D deficiency/insufficiency with obesity, Asian, Chinese and black ethnicity, smoking, never drinking statistically significant? It is unfortunate that the relationship between comorbidities (e.g., as in ref 7) and vitamin D status was not investigated.

Response:

We appreciate the reviewer's comment on the absence of statistical analysis. We further analyzed the association between demographic factors and vitamin D deficiency or insufficiency using simple and

multivariable logistic regression models. We have revised the methods, results and the discussion based on our new analysis. The results were shown in Table 3:

Table 3 The association between demographic characteristics and low vitamin D status

Characteristics	Odds ratio of vitamin D insufficiency (25(OH)D < 50 nmol/L)		Odds ratio of vitamin D deficiency (25(OH)D < 25 nmol/L)	
	Crude	Adjusted ¹	Crude	Adjusted ¹
Sex				
Female	1	1	1	1
Male	1.02 (1.0, 1.03)	0.91 (0.90, 0.93)	1.02 (1.01, 1.04)	0.91 (0.9, 0.93)
Age (SD)	0.98 (0.97, 0.97)	0.98 (0.98, 0.98)	0.97 (0.96, 0.97)	0.98 (0.98, 0.98)
BMI				
Healthy weight	1	1	1	1
Underweight	1.3 (1.20, 1.42)	1.26 (1.14, 1.38)	1.83 (1.64, 2.03)	1.71 (1.51, 1.93)
Overweight	1.25 (1.23, 1.26)	1.24 (1.22, 1.25)	1.11 (1.09, 1.13)	1.04 (1.01, 1.06)
Obese	2.15 (2.11, 2.18)	2.08 (2.04, 2.12)	1.91 (1.86, 1.95)	1.68 (1.64, 1.72)
Ethnic background				
White	1	1	1	1
Mixed	2.30 (2.11, 2.51)	2.24 (2.03, 2.46)	2.42 (2.21, 2.64)	2.31 (2.09, 2.56)
Asian	8.67 (8.02, 9.35)	8.54 (7.87, 9.27)	8.47 (8.10, 8.86)	10.99 (10.39, 11.62)
Black	4.74 (4.44, 5.06)	4.14 (3.85, 4.45)	3.93 (3.74, 4.13)	3.6 (3.39, 3.83)
Chinese	4.25 (3.70, 4.89)	4.42 (3.81, 5.14)	2.72 (2.42, 3.06)	2.77 (2.42, 3.18)
Other ethnicity	2.91 (2.74, 3.10)	2.73 (2.55, 2.93)	3.24 (3.07, 3.42)	3.11 (2.9, 3.33)
Tobacco smoking				
Non-smoker	1	1	1	1
Current smoker	1.53 (1.50, 1.56)	1.43 (1.40, 1.46)	1.94 (1.90, 1.99)	1.82 (1.77, 1.87)
Alcohol drinking status				
Never drink	1	1	1	1
Drink occasionally	0.78 (0.76, 0.8)	0.85 (0.84, 0.88)	0.63 (0.61, 0.65)	0.75 (0.73, 0.78)
Drink weekly	0.55 (0.53, 0.56)	0.66 (0.64, 0.68)	0.40 (0.39, 0.42)	0.55 (0.53, 0.57)
Drink daily	0.53 (0.52, 0.55)	0.70 (0.68, 0.72)	0.43 (0.42, 0.45)	0.66 (0.64, 0.69)
Vitamin and mineral supplementation				
Other vitamin and mineral supplement	1	1	1	1

Vitamin D, multivitamin, fish oil and calcium supplement	0.46 (0.45, 0.47)	0.41 (0.41, 0.42)	0.36 (0.35, 0.37)	0.32 (0.31, 0.33)
Index of multiple deprivation (IMD)				
1 (Least deprived)	1	1	1	1
2	1.11 (1.09, 1.13)	1.03 (1.01, 1.05)	1.12 (1.09, 1.16)	1.02 (0.98, 1.05)
3	1.18 (1.16, 1.2)	1.05 (1.03, 1.07)	1.29 (1.26, 1.33)	1.11 (1.07, 1.14)
4	1.44 (1.42, 1.47)	1.17 (1.14, 1.19)	1.69 (1.64, 1.74)	1.27 (1.23, 1.31)
5 (Most deprived)	1.91 (1.87, 1.94)	1.34 (1.31, 1.37)	2.43 (2.36, 2.50)	1.53 (1.48, 1.58)
Seasons				
Summer	1	1	1	1
Spring	3.48 (3.42, 3.54)	3.86 (3.79, 3.93)	5.26 (5.1, 5.42)	6.43 (6.22, 6.65)
Autumn	1.37 (1.35, 1.40)	1.43 (1.40, 1.45)	1.74 (1.68, 1.80)	1.89 (1.82, 1.96)
Winter	4.06 (3.99, 4.14)	4.56 (4.47, 4.65)	6.31 (6.12, 6.51)	7.82 (7.55, 8.1)
Regions (categorized centres)				
South West	1	1	1	1
South East	1.05 (1.03, 1.08)	1.15 (1.11, 1.18)	1.04 (0.98, 1.09)	1.11 (1.05, 1.18)
London	1.86 (1.81, 1.91)	1.31 (1.28, 1.35)	2.34 (2.24, 2.44)	1.31 (1.25, 1.38)
East Midlands	1.19 (1.15, 1.23)	1.07 (1.04, 1.11)	1.28 (1.22, 1.35)	1.13 (1.07, 1.2)
West Midlands	1.88 (1.82, 1.93)	1.22 (1.18, 1.26)	2.38 (2.27, 2.48)	1.26 (1.19, 1.32)
Wales	1.81 (1.75, 1.88)	1.03 (0.99, 1.07)	1.94 (1.84, 2.05)	1.03 (0.97, 1.09)
Yorkshire and The Humber	1.45 (1.41, 1.48)	1.19 (1.16, 1.23)	1.68 (1.61, 1.76)	1.30 (1.24, 1.36)
North West	1.38 (1.35, 1.42)	1.15 (1.12, 1.18)	1.68 (1.61, 1.76)	1.28 (1.22, 1.34)
North East	1.39 (1.35, 1.43)	1.23 (1.19, 1.27)	1.78 (1.7, 1.86)	1.53 (1.46, 1.61)
Scotland	2.83 (2.74, 2.92)	1.98 (1.91, 2.05)	3.58 (3.43, 3.75)	2.39 (2.28, 2.51)

1. Adjusted for sex, age, BMI categories, ethnicity, smoking, drinking, vitamin D supplementation, index of multiple deprivation, seasons, regions of UK Biobank assessment centres; 3. IMD scores were by quintile.

Reviewer 1 / comment 7

In the Discussion, it is claimed that a larger sample size gives a "more accurate distribution of vitamin D status". Obviously this assertion is subject to inherent biases and methodological issues.

Response:

We agree with the reviewer that having large samples cannot guarantee that we can describe the distribution of vitamin D status in the UK population accurately. Nevertheless, the large sample size provides more statistical power for assessing the association, and our study is still internally valid. We have revised the discussion section to address this issue in detail:

However, several important limitations need to be considered. Despite the large size of the UK Biobank, this cohort is not nationally representative. Compared with non-participants, the participants of the UK Biobank were more likely to be older, female, and with a higher socioeconomic status and fewer health conditions¹². Due to this healthy volunteer effect, our findings on vitamin D status cannot be generalised to the UK population, and we may have underestimated the prevalence of vitamin D deficiency. Nevertheless, as the healthy volunteer effect does not affect the validity of exposure–outcome relationships, our findings on the factors associated with vitamin D status are likely to be valid.

Reviewer 2 / comment 1

The serum 25D measurements were carried out using a chemiluminescence system that is no longer considered to be the gold-standard for analysis of 25D (liquid chromatography- tandem mass spectrometry) is now accepted as the definitive method for accurate vitamin D analysis) The author must provide detailed information on the reproducibility – intra-assay variation and DEQAS data for the assay that was used.

Response:

We appreciate the reviewer’s comment on the vitamin D measurement. In the UK, both chemiluminescence immunoassay (CLIA) or tandem mass spectrometry, which are recognized by Royal Osteoporosis Society and Public Health England^{6,7}. We have now added further information on the reproducibility and other measures of quality assurance for the assay. The analytic machine used in this study was certified by CDC’s Vitamin D Standardization-Certification Program (VDSCP), and the laboratory procedures were examined by an external quality assurance scheme.

We have changed the sentence in the method section as below:

Serum 25-hydroxyvitamin D status was measured by chemiluminescence immunoassay (DiaSorin Ltd. LIASON XL, Italy), which was certified by the Vitamin D Standardization-Certification Program of the Centers for Disease Control and Prevention¹⁵. To ensure the precision of analysis, quality control samples at different concentrations were analysed¹⁶, and the testing assay for vitamin D was verified through the RIQAS Immunoassay Speciality I EQA programme (Randox Laboratories Ltd.), an external quality assurance scheme¹⁷

Reviewer 2 / comment 2

There appear to be some interesting variations in Table 2. For example, obese people are far less likely to be vitamin D-sufficient. Likewise, daily drinkers are far more likely to be vitamin D-sufficient but is this because this includes more of the Caucasian cohort? Trends need to be statistically defined – e.g. how much more time outdoors is required to produce vitamin D-sufficiency. At present the study is simply a list of data, there is no attempt to analyse these data.

Response:

We appreciated reviewer’s comment on our analysis. We further analyzed the association between demographic factors and vitamin D deficiency or insufficiency using crude and multivariable adjusted logistic regression models. We have revised the method, results and the discussion based on our new analysis. The results were shown in Table 3. We further discussed our finding about the association between alcohol drinking and vitamin D status in our discussion section. Regarding outdoor time and vitamin D, we agree that this is an important area that requires further research. However, this question is beyond the scope of this study.

Table 3 The association between demographic characteristics and low vitamin D status

Characteristics	Odds ratio of vitamin D insufficiency (25(OH)D < 50 nmol/L)		Odds ratio of vitamin D deficiency (25(OH)D < 25 nmol/L)	
	Crude	Adjusted ¹	Crude	Adjusted ¹
Sex				
Female	1	1	1	1

Male	1.02 (1.0, 1.03)	0.91 (0.90, 0.93)	1.02 (1.01, 1.04)	0.91 (0.9, 0.93)
Age (SD)	0.98 (0.97, 0.97)	0.98 (0.98, 0.98)	0.97 (0.96, 0.97)	0.98 (0.98, 0.98)
BMI				
Healthy weight	1	1	1	1
Underweight	1.3 (1.20, 1.42)	1.26 (1.14, 1.38)	1.83 (1.64, 2.03)	1.71 (1.51, 1.93)
Overweight	1.25 (1.23, 1.26)	1.24 (1.22, 1.25)	1.11 (1.09, 1.13)	1.04 (1.01, 1.06)
Obese	2.15 (2.11, 2.18)	2.08 (2.04, 2.12)	1.91 (1.86, 1.95)	1.68 (1.64, 1.72)
Ethnic background				
White	1	1	1	1
Mixed	2.30 (2.11, 2.51)	2.24 (2.03, 2.46)	2.42 (2.21, 2.64)	2.31 (2.09, 2.56)
Asian	8.67 (8.02, 9.35)	8.54 (7.87, 9.27)	8.47 (8.10, 8.86)	10.99 (10.39, 11.62)
Black	4.74 (4.44, 5.06)	4.14 (3.85, 4.45)	3.93 (3.74, 4.13)	3.6 (3.39, 3.83)
Chinese	4.25 (3.70, 4.89)	4.42 (3.81, 5.14)	2.72 (2.42, 3.06)	2.77 (2.42, 3.18)
Other ethnicity	2.91 (2.74, 3.10)	2.73 (2.55, 2.93)	3.24 (3.07, 3.42)	3.11 (2.9, 3.33)
Tobacco smoking				
Non-smoker	1	1	1	1
Current smoker	1.53 (1.50, 1.56)	1.43 (1.40, 1.46)	1.94 (1.90, 1.99)	1.82 (1.77, 1.87)
Alcohol drinking status				
Never drink	1	1	1	1
Drink occasionally	0.78 (0.76, 0.8)	0.85 (0.84, 0.88)	0.63 (0.61, 0.65)	0.75 (0.73, 0.78)
Drink weekly	0.55 (0.53, 0.56)	0.66 (0.64, 0.68)	0.40 (0.39, 0.42)	0.55 (0.53, 0.57)
Drink daily	0.53 (0.52, 0.55)	0.70 (0.68, 0.72)	0.43 (0.42, 0.45)	0.66 (0.64, 0.69)
Vitamin and mineral supplementation				
Other vitamin and mineral supplement	1	1	1	1
Vitamin D, multivitamin, fish oil and calcium supplement	0.46 (0.45, 0.47)	0.41 (0.41, 0.42)	0.36 (0.35, 0.37)	0.32 (0.31, 0.33)
Index of multiple deprivation (IMD)				
1 (Least deprived)	1	1	1	1
2	1.11 (1.09, 1.13)	1.03 (1.01, 1.05)	1.12 (1.09, 1.16)	1.02 (0.98, 1.05)
3	1.18 (1.16, 1.2)	1.05 (1.03, 1.07)	1.29 (1.26, 1.33)	1.11 (1.07, 1.14)
4	1.44 (1.42, 1.47)	1.17 (1.14, 1.19)	1.69 (1.64, 1.74)	1.27 (1.23, 1.31)
5 (Most deprived)	1.91 (1.87, 1.94)	1.34 (1.31, 1.37)	2.43 (2.36, 2.50)	1.53 (1.48, 1.58)

Seasons				
Summer	1	1	1	1
Spring	3.48 (3.42, 3.54)	3.86 (3.79, 3.93)	5.26 (5.1, 5.42)	6.43 (6.22, 6.65)
Autumn	1.37 (1.35, 1.40)	1.43 (1.40, 1.45)	1.74 (1.68, 1.80)	1.89 (1.82, 1.96)
Winter	4.06 (3.99, 4.14)	4.56 (4.47, 4.65)	6.31 (6.12, 6.51)	7.82 (7.55, 8.1)
Regions (categorized centres)				
South West	1	1	1	1
South East	1.05 (1.03, 1.08)	1.15 (1.11, 1.18)	1.04 (0.98, 1.09)	1.11 (1.05, 1.18)
London	1.86 (1.81, 1.91)	1.31 (1.28, 1.35)	2.34 (2.24, 2.44)	1.31 (1.25, 1.38)
East Midlands	1.19 (1.15, 1.23)	1.07 (1.04, 1.11)	1.28 (1.22, 1.35)	1.13 (1.07, 1.2)
West Midlands	1.88 (1.82, 1.93)	1.22 (1.18, 1.26)	2.38 (2.27, 2.48)	1.26 (1.19, 1.32)
Wales	1.81 (1.75, 1.88)	1.03 (0.99, 1.07)	1.94 (1.84, 2.05)	1.03 (0.97, 1.09)
Yorkshire and The Humber	1.45 (1.41, 1.48)	1.19 (1.16, 1.23)	1.68 (1.61, 1.76)	1.30 (1.24, 1.36)
North West	1.38 (1.35, 1.42)	1.15 (1.12, 1.18)	1.68 (1.61, 1.76)	1.28 (1.22, 1.34)
North East	1.39 (1.35, 1.43)	1.23 (1.19, 1.27)	1.78 (1.7, 1.86)	1.53 (1.46, 1.61)
Scotland	2.83 (2.74, 2.92)	1.98 (1.91, 2.05)	3.58 (3.43, 3.75)	2.39 (2.28, 2.51)

1. Adjusted for sex, age, BMI categories, ethnicity, smoking, drinking, vitamin D supplementation, index of multiple deprivation, seasons, regions of UK Biobank assessment centres; 3. IMD scores were by quintile.

RESULTS

Association between demographic factors and vitamin D status

To assess the association of factors related to vitamin D status, the odds ratio of having vitamin D deficiency (25(OH)D <25 nmol/L) or insufficiency (25(OH)D <50 nmol/L) were summarised in Table 3. Male sex, abnormal BMI, smoking, non-white ethnicity, and being more deprived had greater odds of vitamin D deficiency or insufficiency. Older age, drinking alcohol, and taking vitamin D, multivitamin, fish oil or calcium supplements were associated with lower odds of vitamin D deficiency or insufficiency. Compared with summer, receiving testing in spring or winter had greater odds of vitamin D deficiency or insufficiency. For the regions with assessment centres, Scotland had higher odds of vitamin D deficiency and insufficiency compared with other regions in the UK. After adjusting for the variables listed in Table 3, these associations remained (Table 3).

DISCUSSION

Curiously, age and drinking alcohol were inversely associated with vitamin D status. Aging has been regarded as a risk factor for vitamin D deficiency; however, our results showed older age slightly decreased the odds of deficiency. A possible explanation for this is that our study population focused on middle-aged or older participants, who may be more likely to receive prescriptions containing vitamin D. However, this information was not included in our study. Regarding drinking alcohol, although a previous systematic review reported no consistent correlation between alcohol intake and serum vitamin D levels²⁷, recent studies have shown a negative association. A study using the National Health and Nutrition Examination Survey (NHANES) database in the US from 2001 to 2010 showed the prevalence of vitamin D deficiency among current alcohol drinkers was 38% lower than in

non-drinkers²⁶. Similarly, a cross-sectional survey in Portugal among 1,500 participants aged over 65 years showed that alcohol drinkers had lower odds of vitamin D deficiency compared with non-drinkers (moderate drinker: OR=0.49 [CI: 0.32 to 0.73]; excessive drinker: OR=0.48 [CI: 0.27 to 0.85])²⁵. The mechanism behind this possible association is still unclear, and future studies about vitamin D should take alcohol into account.

Reviewer 2 / comment 3

The authors show seasonal variations in 25D levels but this is shown in a very user-unfriendly fashion. Most of the UK map is white and not part of the study. Why not just list the locations (maybe South to North or vice versa) for each season and show the actual serum 25 levels?

Response:

We acknowledge that our map might not have been clear, so we listed the vitamin D status based on the different regions the assessment centres were located in in Table 1 as follows. The original map has been moved to supplementary material following the reviewer's comment.

Characteristics	Vitamin D Deficiency (< 25 nmol/L)	Vitamin D Insufficiency (25 – 50 nmol/L)	Vitamin D Sufficiency (> 50 nmol/L)
Regions of UK Biobank assessment centres			
South West (N= 38,872)	3,068 (7.9%)	14,622 (37.6%)	21,182 (54.5%)
South East (N= 39,814)	3,245 (8.2%)	15,400 (38.7%)	21,169 (53.2%)
London (N= 61,291)	10,232 (16.7%)	27,037 (44.1%)	24,022 (39.2%)
East Midlands (N= 30,337)	3,001 (9.9%)	12,115 (39.9%)	15,221 (50.2%)
West Midlands (N= 40,044)	6,785 (17.0%)	17,670 (44.1%)	15,589 (38.9%)
Wales (N= 19,142)	2,732 (14.3%)	8,808 (46.0%)	7,614 (39.8%)
Yorkshire and The Humber (N= 66,197)	8,372 (12.7%)	27,878 (42.1%)	29,947 (45.2%)
North West (N= 68,196)	8,715 (12.8%)	27,924 (41.0%)	31,557 (46.3%)
North East (N=52,277)	6,919 (13.2%)	21,174(40.5%)	24,184 (46.3%)
Scotland (N=32,419)	7,618 (23.5%)	15,163 (46.8%)	9,638 (29.7%)

Reviewer 2 / comment 4

'olive' coloured skin appears to be more commonly associated with vitamin D-sufficiency. Can the authors comment on this? What are the other known features of this group.

Response:

While interesting, we had some concerns about the validity of the skin colour variable. The information about skin colours was obtained through self-report and did not appear to correlate well with ethnicity. As can be seen in the table, some white people reported they had "dark skin," and some black people reported that they had "light skin", which indicated that skin colours may have been very subjective. After discussing with coauthors, we did not include this variable in our analyses.

Table 1 The distribution of ethnicity among different self-reported skin colours

	Light skin
White	369,799 (79.3%)
Mixed	339 (11.8%)
Asian	1,539 (16.2%)
Black	275 (3.5%)
Chinese	397 (29.3%)
Other ethnicity	1,793 (31.3%)

Reviewer 2 / comment 5

Are any data for exercise levels available for the Biobank cohort?

Response:

We agree that this is an important factor in vitamin D. However, our study mainly focused on the effects of seasons, regions and supplementation on vitamin D status, and individual physical activity might be outside the scope of this project. Therefore, we did not include this variable in this analysis.

References

1. Holick MF. Vitamin D deficiency. The New England journal of medicine 2007;357(3):266-81. doi: 10.1056/NEJMra070553 [published Online First: 2007/07/20]
2. Public Health England. PHE publishes new advice on vitamin D 2016 [Available from: <https://www.gov.uk/government/news/phe-publishes-new-advice-on-vitamin-d> accessed 05 October 2020.
3. Sassi F, Tamone C, D'Amelio P. Vitamin D: Nutrient, Hormone, and Immunomodulator. Nutrients 2018;10(11) doi: 10.3390/nu10111656 [published Online First: 2018/11/08]
4. Mann EH, Pfeffer PE, Hawrylowicz CM. Chapter 104 - Vitamin D and Adaptive Immunology in Health and Disease. In: Feldman D, ed. Vitamin D (Fourth Edition): Academic Press 2018:937-49.
5. Martineau AR, Jolliffe DA, Hooper RL, et al. Vitamin D supplementation to prevent acute respiratory tract infections: systematic review and meta-analysis of individual participant data. Bmj 2017;356:i6583. doi: <https://dx.doi.org/10.1136/bmj.i6583>
6. The Scientific Advisory Committee on Nutrition. Vitamin D and Health: Public Health England 2016.
7. Royal Osteoporosis Society. Vitamin D and Bone Health: A Practical Clinical Guideline for Patient Management. 2 ed: Royal Osteoporosis Society,, 2018:10.
8. Santos A, Amaral TF, Guerra RS, et al. Vitamin D status and associated factors among Portuguese older adults: results from the Nutrition UP 65 cross-sectional study. BMJ open 2017;7(6):e016123. doi: 10.1136/bmjopen-2017-016123 [published Online First: 2017/06/25]
9. Liu X, Baylin A, Levy PD. Vitamin D deficiency and insufficiency among US adults: prevalence, predictors and clinical implications. Br J Nutr 2018;119(8):928-36. doi: 10.1017/s0007114518000491 [published Online First: 2018/04/13]
10. Fry A, Littlejohns TJ, Sudlow C, et al. Comparison of Sociodemographic and Health-Related Characteristics of UK Biobank Participants With Those of the General Population. American journal of epidemiology 2017;186(9):1026-34. doi: <https://dx.doi.org/10.1093/aje/kwx246>
11. CDC. CDC Vitamin D Standardization-Certification Program (CDC VDSCP), 2020.
12. UK Biobank. Biomarker assay quality procedures: approaches used to minimise systematic and random errors. 1.2 ed, 2019.
13. UK Biobank. Companion document for serum biomarker data. 1 ed, 2019.

VERSION 2 – REVIEW

REVIEWER	Martin Hewison Institute of Metabolism and Systems Research Level 2, Institute of Biomedical Research The University of Birmingham Birmingham B15 2TT UK
REVIEW RETURNED	08-Nov-2020
GENERAL COMMENTS	The authors have produced a comprehensive and well-structured response to the original comments. the

VERSION 2 – AUTHOR RESPONSE

Reviewer: 2

Reviewer Name: Martin Hewison

Institution and Country: Institute of Metabolism and Systems Research

Level 2, Institute of Biomedical Research

The University of Birmingham

Birmingham

B15 2TT

UK

Competing interests 1: None

Comments to the Author

The authors have produced a comprehensive and well-structured response to the original comments.

We thank the reviewer for these positive comments.